# Identification of Secondary Biomechanical Abnormalities in the Lower Limb Joints after Chronic Transtibial Amputation: A Proof-of-Concept Study Using SPM1D Analysis

**DOI:** 10.3390/bioengineering9070293

**Published:** 2022-06-30

**Authors:** Amr Alhossary, Wei Tech Ang, Karen Sui Geok Chua, Matthew Rong Jie Tay, Poo Lee Ong, Tsurayuki Murakami, Tabitha Quake, Trevor Binedell, Seng Kwee Wee, Min Wee Phua, Yong Jia Wei, Cyril John Donnelly

**Affiliations:** 1Rehabilitation Research Institute of Singapore-Nanyang Technological University, Singapore 308232, Singapore; aalhossary@ntu.edu.sg (A.A.); wtang@ntu.edu.sg (W.T.A.); jiawei.yong@ntu.edu.sg (Y.J.W.); 2Centre of Rehabilitation Excellence, Tan Tock Seng Hospital, Singapore 569766, Singapore; karen_chua@ttsh.com.sg (K.S.G.C.); matthew_rj_tay@ttsh.com.sg (M.R.J.T.); poolee_ong@ttsh.com.sg (P.L.O.); tsurayuki_murakami@ttsh.com.sg (T.M.); tabitha_zh_quake@ttsh.com.sg (T.Q.); trevor_binedell@ttsh.com.sg (T.B.); seng_kwee_wee@ttsh.com.sg (S.K.W.); min_wee_phua@ttsh.com.sg (M.W.P.)

**Keywords:** statistical parametric mapping, MovementRx, transtibial amputation (TTA), clinical gait analysis, walking biomechanics, osteoarthritis, knee dynamics

## Abstract

SPM is a statistical method of analysis of time-varying human movement gait signal, depending on the random field theory (RFT). MovementRx is our inhouse-developed decision-support system that depends on SPM1D Python implementation of the SPM (spm1d.org). We present the potential application of MovementRx in the prediction of increased joint forces with the possibility to predispose to osteoarthritis in a sample of post-surgical Transtibial Amputation (TTA) patients who were ambulant in the community. We captured the three-dimensional movement profile of 12 males with TTA and studied them using MovementRx, employing the SPM1D Python library to quantify the deviation(s) they have from our corresponding reference data, using “Hotelling 2” and “T test 2” statistics for the 3D movement vectors of the 3 main lower limb joints (hip, knee, and ankle) and their nine respective components (3 joints × 3 dimensions), respectively. MovementRx results visually demonstrated a clear distinction in the biomechanical recordings between TTA patients and a reference set of normal people (ABILITY data project), and variability within the TTA patients’ group enabled identification of those with an increased risk of developing osteoarthritis in the future. We conclude that MovementRx is a potential tool to detect increased specific joint forces with the ability to identify TTA survivors who may be at risk for osteoarthritis.

## 1. Introduction

Most of the human joints have a range of movement that includes the three dimensions. In 3D movement, everything is relative and linked together. The rotation component of the movement in the knee is too subtle to be noticed, isolated from other components, using the human eye. Nevertheless, it can be studied and isolated using motion capture (MOCAP) techniques. There is gait variability among individuals, and no person walks the same way as another due to individual differences in symmetry, loading forces, kinematics, etc. [1,2,3,4]. One-dimensional statistical parametric mapping (SPM1D) [5] is a Python/MATLAB library that recently introduced the statistical parametric mapping (SPM) statistical method to the biomechanics community. SPM1D has been employed in the analysis of time-varying human movement gait in both normal and pathological conditions (hemiplegic stroke gait, transtibial amputee gait, and knee OA) using biomechanical measurements (kinematics, moments, and EMG) [6,7,8].

Amputation is one of the oldest surgical operations ever performed [9]. The oldest known prosthesis to replace the shape and function of lost limbs or limb parts was the Cairo toe (between 1550 and 700 BC), found in Thebes-West near Luxor in Egypt [10]. Transtibial amputation (TTA) and transfemoral amputation (TFA) are the most commonly performed surgical amputations. The main reasons for amputations are dysvascular disease associated with diabetes mellitus, trauma, or cancer [11]. In Singapore, diabetes mellitus is a major cause of lower limb loss, with a lower extremity amputation rate of 95.0 per 100,000 population in 2017 [12,13].

Unilateral TFA patients are commonly reported to suffer from secondary knee osteoarthritis (OA) in the contralateral limb. Evidence suggests that the secondary OA is due to repetitive high peak loads during the load-acceptance phase of stance (step-to-step inter-limb body weight transfer) of the contralateral limb [14,15]. Similarly, patients with unilateral TTA exert peak vertical ground reaction force (GRF) using the intact (contralateral) limb during the sit-to-stand movement that is 27% more than the residual limb, predisposing them to contralateral limb OA [16,17]. Osteoarthritis is a progressive condition, wherein early detection and management may slow its progression to late stages through various multimodal interventions, such as education, weight management, appropriate walking aid use and shoe wear selection, injury prevention, aerobic and lower limb strengthening exercises, and activity modification [15,18].

Among the traumatic amputee veteran population, the age and average weight-adjusted prevalence ratio of knee pain among transtibial amputees, compared to nonamputees, was 1.3 (95% confidence interval [CI], 0.7–2.1) for the knee of the contralateral limb and 0.2 (95% CI, 0.05–0.7) for the residual limb knee joint [15]. The standardized prevalence ratio of knee pain in the contralateral limb and symptomatic OA among transfemoral amputees, compared to nonamputees, was 3.3 (95% CI, 1.5–6.3) and 1.3 (95% CI, 0.2–4.8), respectively.

Stresses on the contralateral knee of amputees may contribute to secondary disability [15]. Ferris and her colleagues studied the difference between Ertl and non-Ertl transtibial amputees [19] in the five-times sit-to-stand task and found that there are biomechanical differences between the two categories, and between the two limbs within each category. These differences were not detected by visual inspection but could be measured using the force plate and represented as a similarity index of the normalized peak GRF. Even the more symmetric group (Ertl group) had a difference of 17% between the residual and contralateral limbs [20]. In another study, Melzer and colleagues studied amputees who play volleyball and compared them to non-volleyball players and a control group. They concluded that the amputees have a prevalence of OA in the contralateral knee that is roughly 65% higher than the healthy controls. The most common findings among the amputees were patellar and medial osteophytosis of the tibiofemoral joint, with a tendency to medial narrowing of the tibiofemoral joint space [21]. The greater vertical GRFs thereof may contribute to the increased prevalence of OA in the contralateral limb [22].

Regardless of the unilateral amputation site, whenever there is amputation on one side, there is a tendency for joint disease in large weight-bearing joints above the level of the amputation site and/or in the contralateral side joints [23,24]. We hypothesize that using biomechanical tools we can detect increased joint forces potentially contributing to the development of joint osteoarthritis.

The objective of the study is to perform an exploratory analysis of MOCAP for TTA using SPM, aiming at determining whether biomechanical signals predisposing individuals to contralateral osteoarthritis of large joints can be detected.

## 2. Materials and Methods

### 2.1. Reference and Subjects

The reference dataset of this study contained records of a local normative reference dataset which was collected in the same laboratory [25]. The data came from 20 male subjects, all of which were in the age group 50–70 years and were of Asian ethnicity.

The pool of subjects consisted of 15 TTA patients (12 males and 3 females) who were recruited from a single center, the Tan Tock Seng Hospital Foot Care and Limb Design Center. We excluded the three females to rule out gender-related bias. The 12 remaining male subjects underwent MOCAP gait analysis at the Rehabilitation Research Institute of Singapore, Nanyang Technological University. In a “10 m walk” study, we studied their gait profile, three years after they had undergone the amputation surgery and prothesis fitting. They matched the reference in gender, age group (50–70 years), and ethnicity. The subjects were at least K-level 3 ambulators [26]. Eight of the 12 (66.67%) had right sided TTA and all were independent community ambulators without aids. Visual inspection of their gait prior to MOCAP classified each of the participant’s gait as normal with below-knee prosthesis use. For more information about the characteristics of the subjects, see Appendix A.

#### 2.1.1. Ethics Approval

Ethics approval was granted by the National Healthcare Group Institutional Review Boards (NHG DSRB 2019/00879) and the study was registered under www.clinicaltrials.gov (accessed on 1 January 2022) (NCT 04169594). All subjects gave written informed consent prior to MOCAP procedures.

#### 2.1.2. Patient and Public Involvement

Although the study included TTA patients and normal reference (control) persons as subjects, none of them were involved in the design, conduct, research question formulation, choice of outcome measures, reporting, or dissemination plans.

### 2.2. The Study Processes

To ensure the validity of any results produced by our system and that any detected deviations were not merely the outcome of individual variation, we started by studying the variability within normal (control) persons, between normal persons and patients, and between the residual (amputated) and contralateral (healthy) limbs within the patients set.

#### 2.2.1. Control Group Biomechanics (Collectively)

The 20 records of the reference dataset were used to study the variability within the normal persons—being considered “normal”—by leave-one-out (LOO) analysis. That was achieved by choosing one of the 20 records, taking it off the set, studying his deviation from the remaining normal 19 persons, returning it, then repeating the same procedure until all the 20 persons were studied.

#### 2.2.2. Patients’ Biomechanics (Individually)

MovementRx [27] was used to visualize the 12 TTA patients’ data both collectively in an overview graph, and individually one by one, to study the cases in a cascaded manner as needed. MovementRx is a GUI-enabled decision-support application developed in our group. It depends on the SPM1D Python library of random field theory (RFT) statistics to provide a cascaded view of the patient’s chain of joint movement profile from one whole limb to a specific joint, down to a single dimension using one click. The Hotelling 2 test was used to compare the 3D movement vector and the T test 2 was used to compare each dimension components to assess the deviation from the normal reference dataset. The values of measures were visualized using a color-coded view (in the form of a heatmap from blue to red). The color coding was obtained by linear mapping of deviation value from lower limit to higher limit. The details of implementations of MovementRx are explained in another paper, currently under review.

### 2.3. Data Acquisition

Three-dimensional human movement data were captured using sixteen 2 megapixels Miqus M3 and two Miqus video cameras with Qualisys motion capture system (Qualisys, AB, Sweden). Retro-reflective optical markers were placed on the body of a participant according to the marker placement set (see Appendix A). The marker set was based on a modified calibrated anatomical system technique (CAST) [28,29], and markers had a diameter of 12.5 mm (for the body) and 10 mm (for fingertips).

The Qualisys Track Manager (QTM) (Version 2020.2, Qualisys, Sweden) served as the client for data recording. QTM can synchronize the cameras with external devices, such as force plates. Two 600 × 500 × 50 mm Kistler force plates (Type 9260AA6, Kistler, Switzerland) were embedded on the floor to capture ground reaction force during gait. Motion data were captured at 200 Hz and force plate at 2000 Hz synchronously. Figure 1 shows an overview of the motion capture laboratory.

The patient was required to complete at least 6 valid trials, where only one entire foot was set on the pressure plate at a time. Trials where any parts of the foot fell outside the force plate, or more than one foot was placed on the same plate, were considered invalid. The maximum number of valid recorded trials was 13.

### 2.4. Data Processing

Recorded movement data were manually labeled, and gap filled using QTM on all trials. Labelled data were then exported into Visual 3D Professional software (v2021.04.1, C-Motion Inc., Germantown, MD, USA) for further processing. The model was implemented, and pipelines were used for batch processing in Visual 3D. The pipeline started by opening the dynamic trials for batch processing. We interpolated missing data points, followed with the maximum interpolate-able framed gap size set to 10 and number of frames before/after the gap set to 3, using a 3rd-order polynomial function. That was followed by a 4th-order bidirectional low-pass Butterworth filter with the cutoff frequency at 15 Hz, applied to both movement and force data. Hip joint center was estimated using CODA pelvis method provided by Visual 3D [30,31]. The joint coordinate system (JCS) for the reporting of human joint motion is based on recommendation by International Society of Biomechanics (ISB) [32].

Next, the pipeline created a hybrid model base on the first static file of each trial. The next step was to set the subject height and weight. The final step was to calculate each of the model’s segment for kinematics and kinetics. Only the hip joint center was based on anthropometry (see above, for CODA). The rest of the segments were formed using the anatomical marker position directly (or tracking of the rigid body using CAST; explained and cited in Section 2.3 above).

Automatic gait events detection was set to label the events from L/R Heel strike to the next L/R Heel strike for hip, knee, and ankle joint angle and *X*, *Y*, and *Z* axis to each of the joints to export kinematic data. The gait events were determined using the force plate and kinematics of the markers of the foot, based on the ‘automatic gait event’ setting in Visual 3D. The three orthogonal axes in each kinematic and kinetic reference frame were the same. For kinetic data the automatic gait event was from L/R Heel strike to L/R Toe off. Both kinematic and kinetic data were then time-normalized to 101 data points and exported to comma separated values (.csv) TXT file.

## 3. Results

In Figure 2, we overlapped all the 3D vector recordings of the hip joint moments of our reference persons and their respective same joint of the recordings of the TTA patients. The contralateral limb was plotted on the left side, while the residual limbs were plotted on the right side. Most of the control recordings (except for one participant) laid within the first (blue) band (i.e., mild deviation) or even in the grey area (no statistically significant deviation) below the blue band.

On the other hand, the patients’ recordings showed a tendency to distribute above the first band. That is more obvious on the side of the residual limb.

Figure 3 shows the collective overlapped plots of the TTA patients versus the control reference knee joints. The counterparts of Figure 2 and Figure 3 after excluding patient #6 are provided in Appendix A prospectively.

A collective visual representation of the reference control recordings is shown in Figure 4. Similarly, a collective visual representation of the patients’ recordings is shown in Figure 5. In both figures, the persons/patients are presented one per row. There are three joints per row (Hip, Knee, and Ankle). The moments 3D vector of each joint was analyzed using SPM1D and the severity of deviation exhibited in that joint was presented as a color map (from blue for mild, to red for extreme deviation).

Figure 5 shows the moments of the residual limb (regardless of being the left or right side) on the left side, and the contralateral limb on the right side.

Figure 4 is homogenously dominated by grey (no statistically significant deviation) or blue (mild) areas. That is expected, being that they represent the normal population. On the other hand, Figure 5 is dominated on the left side (the residual limb) by red (extreme deviation) areas and on the right side (contralateral limb) by blue and grey areas, with a prevalence of red areas that is less than the left side but more than Figure 4.

Figure 6 shows a comparative view of the contralateral limbs of two of the twelve cases (cases number 1 and 12, highlighted in Figure 5).

The first row in each side of Figure 6 shows the 3D vector of movement profile of one limb. The comparison cascades down from the whole leg (in the first row), down to their three components (in the second row), and finally down to a single dimension of a single joint (the knee joint) in the lower two rows.

Case #1 had minimal deviation on the contralateral limb, while case #12’s recording showed severe changes on the contralateral limb, most obviously in the knee joint.

## 4. Discussion

The presented data show that SPM1D analysis using MovementRx was able to detect excessive pressures in large weight-bearing joints in the residual and contralateral limbs in high level chronic TTA. The findings in this study show detectable differences between normal reference persons (with outliers) and patients with TTA, and between residual and contralateral limbs. (Figure 2, Figure 3, Figure 4 and Figure 5). Our system presented the distinction between those with low and high levels of deviation from normal within the patients.

Comparing Figure 2 to Figure 3, it can be noticed that the control reference group lies in and below the mild band, while the TTA patients’ recordings are dispersed over the four bands. It is noteworthy here that the residual limbs tend to deviate more than the contralateral (intact) limbs, and that the deviation in the hip joint is more than in the knee joint. Appendix A shows that all the cases with extreme deviation have the deviation in the residual limb hip (5 patients) and some of them (3 patients) have extreme deviation in the residual limb knee as well. This is consistent with the current finding that the extreme deviation is found more in the contralateral hip than knee.

Carefully inspecting Figure 2 and Figure 3, it is noticeable that there is one single reference person whose recording deviated greatly from the rest of the reference group. This person is number 6 in Figure 4. It is noteworthy from Figure 4 that this very high moment is manifested in both right and left sides and in both hip and knee, which indicates that symmetry of joint moments is present, which is generally assumed for healthy populations [33]. Variations in joint moments among healthy adults can be influenced by individual cadence and stride lengths [34], and increasing age [35]. As this subject is from the healthy population and he does not manifest any pain during walking, he is assumed to be an outlier with high variation in joint moments.

In contrast, mostly contralateral abnormalities sparing the residual limb were found in two other subjects. Inspecting Figure 5, a remarkable difference between the residual limb (left side) and the contralateral limbs (right side) was detected. For example, excessive pressure on the residual limb was demonstrated in patient #1, as evidenced by red areas in Figure 5 on the left side (all joints). However, his contralateral side indicated similar pressure to the matched control group. On the other hand, patient # 12 showed high pressures (red bands) on the contralateral hip and knee joints, while his ipsilateral knee was similar to matched controls. Another pattern is found in patients # 4 and 7, where there is high pressure on both ipsilateral and contralateral knees. A third and more extreme pattern is found in patient # 5, where huge (compensatory) forces were exerted on his contralateral knee and ankle joints, leaving his residual limb stress close to normal. The variety of abnormalities supports the prior literature, wherein different patterns of stress shifting between the ipsilateral and contralateral limb contributing to secondary osteoarthritis may be attributed to the cumulative effect of gait abnormalities, increased physiologic loads on the contralateral limb knee, and the hopping and stumbling behavior as proposed by Norvell et al. [15]. More differential analysis of deviations is found in Appendix A.

The main finding of the study can be summarized as showing a deviation from normal in the hips more than the knees, in the residual limbs more than the contralateral limbs, and in the TTA patients more than in the reference population. It is known that the patients with amputation in one limb tend to develop osteoarthritis either higher in the weight-bearing kinetic chain of the ipsilateral limb or in the contralateral limb [36,37], and that amputee gait is associated with higher knee extension and abduction torques in the contralateral knee [38]. Having a high level of pressure continuously exerted on the contralateral (healthy) limb, it is possible that patients # 5 and 12 have a high possibility of developing osteoarthritis. Although reference person # 6 does not feel any pain yet, being an outlier who exerts high pressure on his joints, it is predicted that he may possibly be at a higher risk of osteoarthritis as well.

The findings could potentially allow earlier identification of amputee patients with a higher risk of developing OA. These patients would probably go back to the prosthetist for refitting of their prosthesis. The prosthetist could use this to inform/optimize their componentry choice and prosthetic alignment earlier rather than later, to delay the onset/reduce the magnitude of joint degenerative changes.

## 5. Study Limitations

The default upper and lower bounds of the color scale used in this study have been set empirically. However, this is not the most appropriate way. Instead, they need an automatic way of adjustment based on statistical inference from the data and their size. This automatic bound adjustment might need some research later after a long enough period of follow-up of cases and their prognosis. In addition, TTA subjects were not formally screened or clinically examined for complaints of knee pain, physical signs, knee-specific functional scores, nor radiological features of osteoarthritis as part of the study protocol.

## 6. Future Work

SPM1D analysis using MovementRx can potentially be used as a screening test to identify abnormal joint-loading forces prior to the development of symptoms which could herald the onset of osteoarthritis. Correlation with subjects’ complaints of unilateral or bilateral knee/hip pain, functional knee scores, and radiological investigations could shed light on the possibility of coexisting osteoarthritis and its severity. This study used a relatively small sample size, so a larger representative sample is needed for reproducibility and generalization to the larger TTA and transfemoral population as well, and longitudinal studies for the possible progression of OA. A subsequent study 5–10 years later may be conducted to follow up how these patients and their joints are aging.

## 7. Conclusions

In this paper, we present an application of MovementRx for gait and movement analyses in a sample of chronic TTA patients who are at least K-level 3 ambulators, as a step towards early detection of precursors of osteoarthritis. Our data can visually show the distinction between reference normal persons (with outliers) and TTA patients. It can also show the variability within the TTA patients between residual amputated and contralateral limbs, as well as the variability within the contralateral limbs among patients between those with low and high deviation from the reference normal. This distinction could potentially predict the risk of osteoarthritis early enough to prevent it or delay its onset, if possible. If proven to be effective, MovementRx could be used as a tool that models a new osteoarthritis signature that determines abnormal joint pressures, in order to guide preventive clinical protocols.

## Figures and Tables

**Figure 1 bioengineering-09-00293-f001:**
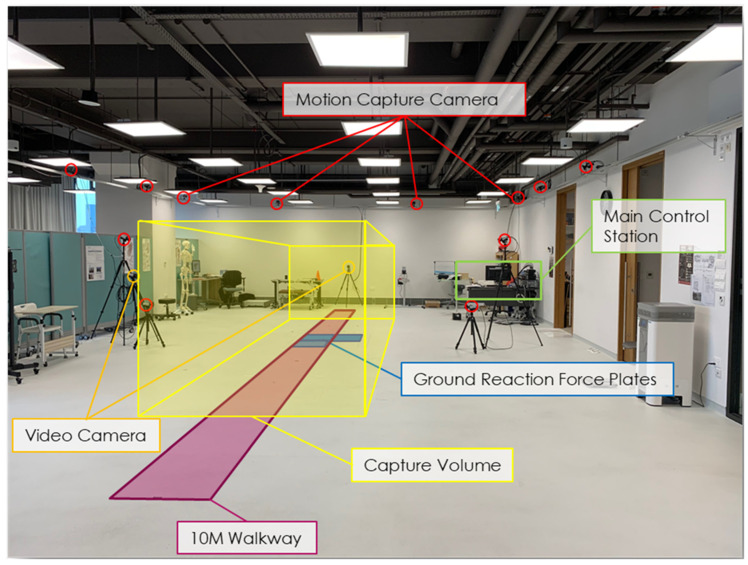
MOCAP laboratory.

**Figure 2 bioengineering-09-00293-f002:**
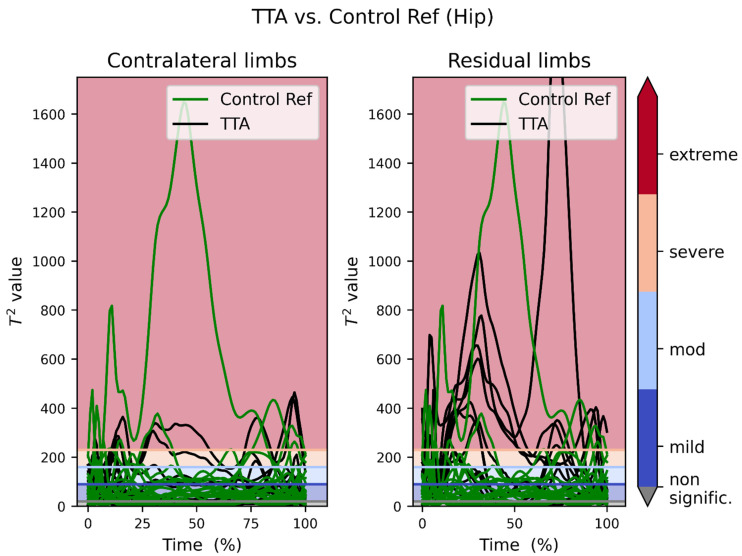
Transtibial amputation patients’ moments recording vs. reference recordings of the hip joint. The color scheme used for the horizontal lines is identical in all figures. The spiky top of one of the curves is truncated for space limitation.

**Figure 3 bioengineering-09-00293-f003:**
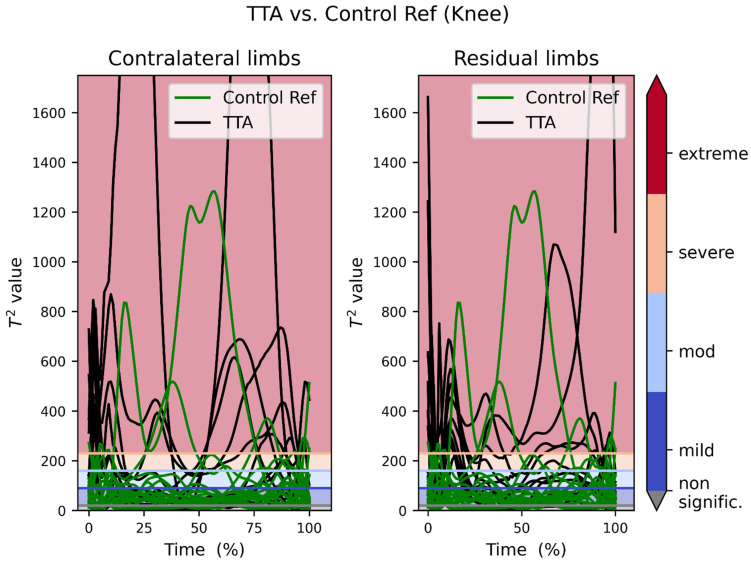
Transtibial amputation patients’ moments recording vs. reference recordings of the knee joint. The color scheme used for the horizontal lines is identical in all figures. The spiky tops of three curves are truncated for space limitation.

**Figure 4 bioengineering-09-00293-f004:**
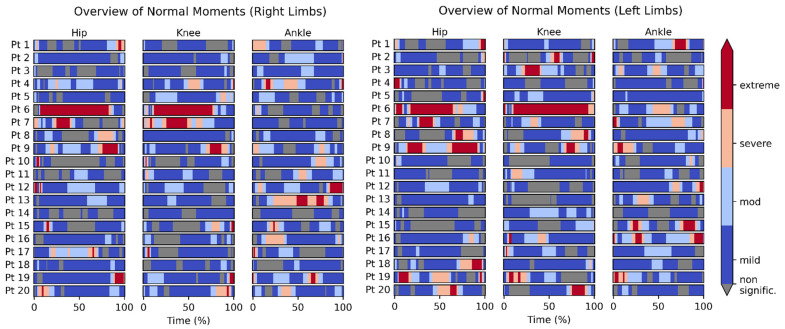
Overview of reference control moments. The color scheme used in this figure is identical in all figures.

**Figure 5 bioengineering-09-00293-f005:**
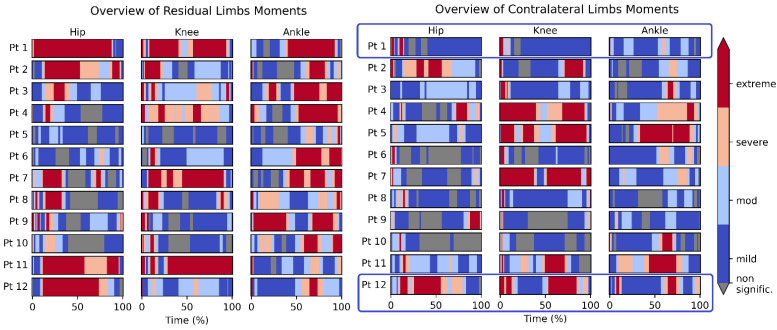
Overview of TTA patients’ moments. The color scheme used in this figure is identical in all figures.

**Figure 6 bioengineering-09-00293-f006:**
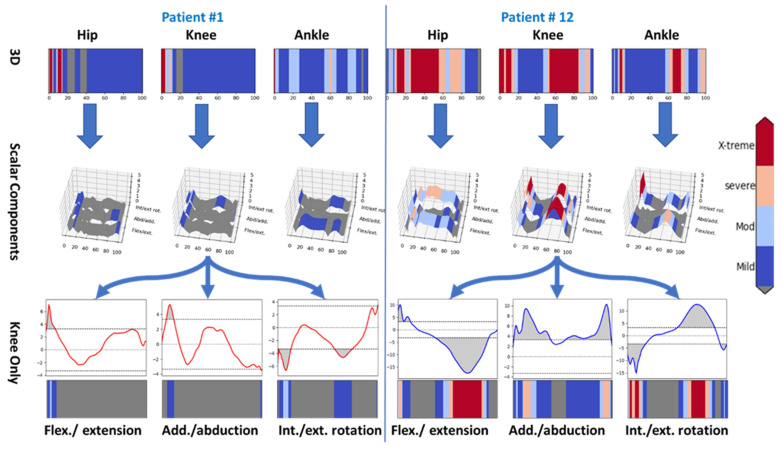
Comparison between two cases with different contralateral limb status. Grey areas represent no significant (subthreshold) effect.

## Data Availability

The aggregated data are provided in the Appendix A. The original individual datasets generated and/or analyzed during the current study are not publicly available due to confidentiality and privacy laws but are available from the corresponding author on reasonable request.

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
