# Peer review of "Identification of Secondary Biomechanical Abnormalities in the Lower Limb Joints after Chronic Transtibial Amputation: A Proof-of-Concept Study Using SPM1D Analysis"

_bioengineering, 2022, doi:10.3390/bioengineering9070293_

Round 1

Reviewer 1 Report

Researchers describe the important topic of amputation. However, the article requires a lot of proofreading before publication.

Introduction

Please put section 1 together (remove all subsections).

Line 37: Please, edit.

Please clearly describe the aim of the study at the end of the Introduction section.

Material and Methods

Include a table with the characteristics of the group in section 2.

Clearly describe the inclusion and exclusion criteria for the group.

Results and discussion

Separate the results and discussion section. In the discussion, refer your observations to the observations of other authors and data from the literature.

Reviewer 2 Report

bioengineering-1636384

Reviewer comments

There are some concerns regarding the establishment of evidence in the Introduction, the validity of the Method and the Discussion of the findings of the study, as listed below:

  • It is stated in the abstract that ‘MovementRx is a versatile gait analysis decision support system that depends on SPM1D python implementation of the Random Field Theory (RFT)’. However, this could be the purpose of the study as most of the relevant information is documented in a paper that is currently under review [L155-156]. Thus, the validity of the methodology used is questionable. Please clarify.
  • Syntax and typing errors should be corrected in parts of the text [i.e. at L12-13; L37-38; L87, L162, L168, etc.].
  • Provide citations for the information presented in (L32-38, L49-51, 85-87, 89-92, etc.). Also, mention the references in the square brackets proximal to the phrase that is related, not at the end of the subsections.
  • Use ‘gait profile’ at L97.
  • Document the exclusion criteria at L102.
  • Use ‘101 data points’ at L140.
  • State which anthropometric model (segmental data) was used in L133-135.
  • Use other terms for ‘normal’ and ‘abnormal’ at L160.
  • L169: see the 1st comment.
  • Place subsection 2.7 prior 2.1.
  • Figures 1+2: scale adequately thus maxima of curves will be depicted.
  • Figure 5: provide SPM statistics for the differences indicated (gray area). Also, indicate the approximate time period within the gait cycle where these differences occur in the respective time domain. In addition, mention the critical value threshold crossing criteria set. If not, provide the rationale for not using the 1-D SPM{i} statistic.
  • It is recommended to re-write subsection 3.4 as section ‘4. Discussion’. In addition, elaborate on the mechanisms that can provide interpretation of the results based on published research evidence.
  • Provide additional reasoning for the limitations of the study.
  • L271-272: see the 1st comment.
  • L283-285: provide this information in subsection 1.6.
  • L286-288: it is rather speculative and it is suggested to be deleted.
  • References: provide all author names.

Reviewer 3 Report

Summary:

Individuals who had undergone trans-tibial amputations were analyzed during their walking gait and compared with matched-controls. A novel software, MovementRx, was used to assess difference between groups.

Comments:

The authors have presented a novel methodology for the assessment of movements in patient populations. There are areas of the manuscript, however, that require additional review and consideration.

The authors do not have a clear hypothesis (see comments below) which drives this research. What is being assessed with these procedures?

The lack of vetting of the controls and the imbalance of number of controls to TTA patients is questionable. Why not have matched numbers, 12 to 12, for comparison? There are no power analyses to justify the random numbers the authors have used. Additionally, were the TTA patients assessed for osteoarthritis prior to gait analysis?

There are other parameters of the gait analysis that were not provided. Such facts as gait speed, number of strides, the length of the walkway, and the number of trials for complete gait analysis were omitted. These factors are necessary in determining the reliability of the data and whether these data can be replicated.

The variables of interest are also ambiguous. The authors mention assessment of the lower limb joints, but there is no mention of the specific joint movements, nor is there mention of the assessment and calculation of moments about the joints for any assessments – although moments are mentioned in the results section.

Abstract: The abstract focuses on the instrumentation of the motion capture system and its capabilities of detecting movement variations. However, the study is really about the detection of movement pattern differences in a population of trans-tibial amputees to compare movement between lower limbs due to the onset of osteoarthritis. This abstract needs to focus on the fundamental component of the study.

Introduction:

Sections 1.1 Human Movement Physiology and 1.2 History – these sections are not necessary as they do not add to the content of the manuscript

Section 1.5 – citations are needed to support these statements

Section 1.6 Research – this section is ambiguous. The hypothesis indicates that differences may be detected. This hypothesis does not provide enough structure to formally state what the driving idea will be for the study. What do the authors believe will be different?

Methods:

Section 2.1 Subjects – the authors state that 12 males with TTA participated, but then controls were also used, but there was no mention of the specifics of the controls. Were the control participants age, height, and body mass-matched with the TTA patients? This needs to be clarified

Section 2.4 Exporting of data – how were he gait events determined (kinematics of the markers of the foot, such as velocity, or a threshold level of the force platforms)? Were the three orthogonal axes the same in each kinematic and kinetic reference frame?

Section 2.5 Reference data – this section needs to be combined with section 2.1. Why are the reference/control participants more than the patient participants?

Section 2.7 – what are “abnormal” patients?

There were no statistical analyses performed, only comparisons of individuals within groups to examine variability within each TTA and control group, respectively. However, the MovementRx was mentioned for movement analysis. What specific movement variables were assessed with this software? Why were the controls not grouped into a single value for each variable?

Results and Discussion:

Lines 174-178: the authors state that the Fig 1 is the hip, but it is not clear what component(s) of the hip movement are illustrated. In addition, the authors did not provide sufficient vetting of the control participant data for a comparison between the controls and TTA patients

Lines 185-186: similar to the hip, what components of the knee joint are represented in Fig 2?

Much of this section is comparing data in controls and TTA patients. There is no reference to the literature and how these data relate to the data that are currently in the literature. The authors need to refrain from discussing the MovementRx as an equivalent to a gene sequencing method as this has no bearing on the current data.

Author Response

Individuals who had undergone trans-tibial amputations were analyzed during their walking gait and compared with matched-controls. A novel software, MovementRx, was used to assess difference between groups.

Comments:

The authors have presented a novel methodology for the assessment of movements in patient populations. There are areas of the manuscript, however, that require additional review and consideration.

  • The authors do not have a clear hypothesis (see comments below) which drives this research. What is being assessed with these procedures?

Thank you for pointing out that. The manuscript had undergone major rewriting including the hypothesis and objective.

  • The lack of vetting of the controls and the imbalance of number of controls to TTA patients is questionable. Why not have matched numbers, 12 to 12, for comparison? There are no power analyses to justify the random numbers the authors have used. Additionally, were the TTA patients assessed for osteoarthritis prior to gait analysis?

This is not a comparative study that compares a set of patients to an equal size of control group. Instead, each patient is compared (individually) to the reference set of matching gender, age, and ethnicity.

Statistical parametric mapping (SPM) is a statistical method that uses Random Field Theory (RFT) [1] to perform topological inference on time series data which is preferable to visual subjective interpretations of the time series – common clinical gait analysis practice today. SPM can be applied to, any spatiotemporally registered and smooth signal. SPM can detect any curve differences (e.g. kinematics, moments, EMG etc.) that change in time [2]. The One-Dimensional Statistical Parametric Mapping (SPM1D) [3] library takes the reference set (where every person is represented as a single record), builds a mathematical model thereof, and then use that model to evaluate the deviation of a new patient (represented as a set of records, one record per trial) or of a test group (represented as one record per patient). Regarding the reference size, it is, actually, the bigger the size of the reference the better, because it gives more accurate model.

The patients were assessed for osteoarthritis prior to gait analysis.

  • There are other parameters of the gait analysis that were not provided. Such facts as gait speed, number of strides, the length of the walkway, and the number of trials for complete gait analysis were omitted. These factors are necessary in determining the reliability of the data and whether these data can be replicated.

Gait speed, length of walkway (10 meters study) and other parameters are available in section 2.2 “The study processes” and in the supplementary data.

The minimum number of trials a patient must have made is not less than 6 good trials: The patient foot is fully set on the pressure plate and not overlapped with any part of the other foot.

  • The variables of interest are also ambiguous. The authors mention assessment of the lower limb joints, but there is no mention of the specific joint movements, nor is there mention of the assessment and calculation of moments about the joints for any assessments – although moments are mentioned in the results section.

We model the movement of each of the three joints (hip, knee, and ankle) as components in the three dimensions (X, Y, and Z) as well as (3D vector) as convention in MovementRx is. I attached the MovementRx manuscript (currently under review) to this revision as an unpublished file.

We model both moments and Kinematics. However, in the study we focus on the moments.

  • Abstract: The abstract focuses on the instrumentation of the motion capture system and its capabilities of detecting movement variations. However, the study is really about the detection of movement pattern differences in a population of trans-tibial amputees to compare movement between lower limbs due to the onset of osteoarthritis. This abstract needs to focus on the fundamental component of the study.

We modified the abstract as suggested.

Introduction:

  • Sections 1.1 Human Movement Physiology and 1.2 History – these sections are not necessary as they do not add to the content of the manuscript

They were reduced as suggested.

  • Section 1.5 – citations are needed to support these statements

Citations were added as suggested.

  • Section 1.6 Research – this section is ambiguous. The hypothesis indicates that differences may be detected. This hypothesis does not provide enough structure to formally state what the driving idea will be for the study. What do the authors believe will be different?

Thank you for pointing that out. This section (paragraph) was rephrased, and an objective was added.

Methods:

  • Section 2.1 Subjects – the authors state that 12 males with TTA participated, but then controls were also used, but there was no mention of the specifics of the controls. Were the control participants age, height, and body mass-matched with the TTA patients? This needs to be clarified

We restructured the manuscript, forming section 2.1. “Reference and Subjects” for that contains the information about both the reference and the subjects. Additional data is available in the supplementary data.

  • Section 2.4 Exporting of data – how were the gait events determined (kinematics of the markers of the foot, such as velocity, or a threshold level of the force platforms)? Were the three orthogonal axes the same in each kinematic and kinetic reference frame?

The gait events were determined using the force plate and kinematics of the markers of the foot, based on the 'automatic gait event' setting in Visual 3D Professional software (v2021.04.1, C-Motion, Germantown, PA, USA). The three orthogonal axes in each kinematic and kinetic reference frame are the same.

  • Section 2.5 Reference data – this section needs to be combined with section 2.1. Why are the reference/control participants more than the patient participants?

Section 2.5 Reference data is combined with 2.1 as suggested.

The reference size point is answered earlier. Please see above.

  • Section 2.7 – what are “abnormal” patients?

This was a bad phrasing of the sentence. It was rephrased to reflect the right meaning and specific terminology was used.

  • There were no statistical analyses performed, only comparisons of individuals within groups to examine variability within each TTA and control group, respectively. However, the MovementRx was mentioned for movement analysis. What specific movement variables were assessed with this software? Why were the controls not grouped into a single value for each variable?

Section 2.7.1 (currently 2.2.1) is just to confirm that when we find a deviation in any of the patients’ recordings, it is most likely to be a deviation not just individual variation. This was achieved by visually showing that the variation among the normal population is less than that among the amputated patients.

Results and Discussion:

  • Lines 174-178: the authors state that the Fig 1 is the hip, but it is not clear what component(s) of the hip movement are illustrated. In addition, the authors did not provide sufficient vetting of the control participant data for a comparison between the controls and TTA patients

Lines 185-186: similar to the hip, what components of the knee joint are represented in Fig 2?

The figures depict the deviation analysis of the moments in the 3Dimensional Vector of hip and knee respectively.

  • Much of this section is comparing data in controls and TTA patients. There is no reference to the literature and how these data relate to the data that are currently in the literature. The authors need to refrain from discussing the MovementRx as an equivalent to a gene sequencing method as this has no bearing on the current data.

We referred our data to the information available in the literature. We also removed the sentence about gene sequencing as advised.

  1. Adler, R.J.; Taylor, J.E. Random fields and geometry; Springer Science & Business Media: 2009.
  2. Serrien, B.; Goossens, M.; Baeyens, J.P. Statistical parametric mapping of biomechanical one-dimensional data with Bayesian inference. Int Biomech 2019, 6, 9-18, doi:10.1080/23335432.2019.1597643.
  3. Pataky, T.C. One-dimensional statistical parametric mapping in Python. Comput Methods Biomech Biomed Engin 2012, 15, 295-301, doi:10.1080/10255842.2010.527837.

Reviewer 4 Report

The authors sought to present the potential application of MovementRx in the early prediction of the risk of developing osteoarthritis in cases of postsurgical Transtibial Amputation. The purposes of this study appear to be specific, but the research novelty and methodology should be further highlighted and elaborated. The authors need to make major revisions to the article, here below are specific suggestions for this study.

  1. Title, it is suggested that the authors should reconsider the title of this study since it may not reflect the main purposes in this version.
  2. Abstract, “MovementRx is a versatile gait analysis decision support system that depends on SPM1D python implementation of the Random Field Theory (RFT). It can analyze the gait against a reference normal and represents the level of data deviation from the reference in a visually intuitive cascaded way”, it is suggested that the authors should briefly highlight the importance of performing this study, rather than barely introduce what MovementRx is capable of.
  3. “we present”, “We captured”, the objective statement is recommended throughout this manuscript.
  4. “3 big lower limb joints”, change to “3 main lower limb joints”.
  5. It is suggested that more result details should be presented in this part.
  6. “We conclude that MovementRx can be used for early prediction of osteoarthritis in cases of surgical Transtibial Amputation.”, the results were not strong enough to support this conclusion since only 12 subjects were involved in this study. Further research with a larger sample size is warranted.
  7. Please modify and improve the quality of the keywords as this will assist others when they are searching for information on your research topic.
  8. Introduction, although the introduction has explained the main purpose of this study, it is suggested that the novelty should be further highlighted.
  9. 1-1.6, sub-sessions are not necessary here, it is suggested that the Introduction should be reorganized with stronger logic.
  10. “There is gait variability among individuals, and no person walks the same way as the other. there are individual differences in symmetry, loading forces. Kinematics etc.”, in terms of gait variability, recent related literature should be cited for further demonstration.
  • Effects of limb dominance on the symmetrical distribution of plantar loading during walking and running. Proceedings of the Institution of Mechanical Engineers, Part P: Journal of Sports Engineering and Technology, 2020, 1754337120960962.
  • Gait variability and symmetry remain consistent during high-intensity 10,000 m treadmill running. Journal of biomechanics, 2018, 79, 129-134.
  • A current review of foot disorder and plantar pressure alternation in the elderly. Physical Activity and Health, 2020, 4(1), 95-106.
  • Gait variability as digital biomarker of disease severity in Huntington’s disease. Journal of neurology, 2020, 267(6), 1594-1601.
  1. “the major lower extremity amputation rate has increased from 11.0 per 100,000 population in 2008 to 13.3 per 100,000 population in 2013 [3, 4].”, update the number information if possible.
  2. It is suggested that a clear research purpose should be given at the end of this part.
  3. Materials and methods, the sample size would affect the result of statistical analysis. The authors should illustrate whether the sample size of this trial was calculated or just be set according to experience?
  4. What are the inclusion criteria for subject recruitment?
  5. What specific motion did the author study in this research? Please specify.
  6. “Data acquisition”, it is suggested that a detailed figure about the experiment setup can help readers understand the content more intuitively.
  7. “Data processing”, it is not necessary to detail each step for data processing in Visual 3D.
  8. 3-2.5, it is suggested that the author could consider merging these sub-sessions.
  9. Results and discussion, both Results and Discussion should be separated parts, and in Results part the authors should present the main findings of this study while in Discussion part the authors should concentrate on the comparisons with previous related studies and further reveal the underlying mechanisms behind their findings.
  10. Conclusion, this part should also be further strengthened based on the main findings of this study.

Author Response

  • The authors sought to present the potential application of MovementRx in the early prediction of the risk of developing osteoarthritis in cases of postsurgical Transtibial Amputation. The purposes of this study appear to be specific, but the research novelty and methodology should be further highlighted and elaborated. The authors need to make major revisions to the article, here below are specific suggestions for this study.

    1. Title, it is suggested that the authors should reconsider the title of this study since it may not reflect the main purposes in this version.

    We agree that mentioning both ipsilateral and contralateral sides is a redundancy. Therefore, we removed it. We also changed the wording. In case you You still propose something different, please advise.

    1. Abstract, “MovementRx is a versatile gait analysis decision support system that depends on SPM1D python implementation of the Random Field Theory (RFT). It can analyze the gait against a reference normal and represents the level of data deviation from the reference in a visually intuitive cascaded way”, it is suggested that the authors should briefly highlight the importance of performing this study, rather than barely introduce what MovementRx is capable of.

    Thank you for the suggestion. After reviewing the abstract, we agree with you that we should focus on what this study is about rather than talking about MovementRx. Therefore, we started the abstract with 1 line about SPM instead, followed directly by talking about this study, and including one line about the usage of MovementRx as our inhouse developed GUI application.

    1. “we present”, “We captured”, the objective statement is recommended throughout this manuscript.

    Thank you for your suggestion. We revised the manuscript as suggested.

    1. “3 big lower limb joints”, change to “3 main lower limb joints”.

    Thank you for the suggestion. It was modified as advised.

    1. It is suggested that more result details should be presented in this part.

    Thank you for the suggestion. We added more results details as advised.

    1. “We conclude that MovementRx can be used for early prediction of osteoarthritis in cases of surgical Transtibial Amputation.”, the results were not strong enough to support this conclusion since only 12 subjects were involved in this study. Further research with a larger sample size is warranted.

    We agree with the reviewer that the sample size is a little tight. Unfortunately, due to logistic and legal issue related to agreements with our clinical partners, this sample is the maximal size we can have in a small country like Singapore. However, we modified our conclusion -as well as other places- to add a sentence about the sample size here and in the study limitation for this reason.

    1. Please modify and improve the quality of the keywords as this will assist others when they are searching for information on your research topic.

    Thank you for pointing out this. We modified the keywords accordingly.

    1. Introduction, although the introduction has explained the main purpose of this study, it is suggested that the novelty should be further highlighted.

    After reorganizing the introduction, the importance of the manuscript is more obvious in the last 2 paragraphs. Additionally, we added 1 more line to the last paragraph to highlight the current innovation.

    1. 1-1.6, sub-sessions are not necessary here, it is suggested that the Introduction should be reorganized with stronger logic.

    Thank you for your recommendation. We reorganized the introduction as suggested.

    1. “There is gait variability among individuals, and no person walks the same way as the other. there are individual differences in symmetry, loading forces. Kinematics etc.”, in terms of gait variability, recent related literature should be cited for further demonstration.
    • Effects of limb dominance on the symmetrical distribution of plantar loading during walking and running. Proceedings of the Institution of Mechanical Engineers, Part P: Journal of Sports Engineering and Technology, 2020, 1754337120960962.
    • Gait variability and symmetry remain consistent during high-intensity 10,000 m treadmill running. Journal of biomechanics, 2018, 79, 129-134.
    • A current review of foot disorder and plantar pressure alternation in the elderly. Physical Activity and Health, 2020, 4(1), 95-106.
    • Gait variability as digital biomarker of disease severity in Huntington’s disease. Journal of neurology, 2020, 267(6), 1594-1601.

    Thank you for the recommendation, we added the references as suggested.

    1. “the major lower extremity amputation rate has increased from 11.0 per 100,000 population in 2008 to 13.3 per 100,000 population in 2013 [3, 4].”, update the number information if possible.

    We updated the numbers using a more recent reference.

    1. It is suggested that a clear research purpose should be given at the end of this part.

    Thank you for the recommendation. We added a research purpose statement as suggested.

    1. Materials and methods, the sample size would affect the result of statistical analysis. The authors should illustrate whether the sample size of this trial was calculated or just be set according to experience?

    The sample size is restricted to all available subjects recruited by our clinical partner.

    1. What are the inclusion criteria for subject recruitment?

    The inclusion criteria are now mentioned in section 2.1. “Reference and Subjects”.

    1. What specific motion did the author study in this research? Please specify.\

    The study was “10 meters walk”. It was explicitly added to the manuscript.

    1. “Data acquisition”, it is suggested that a detailed figure about the experiment setup can help readers understand the content more intuitively.

    Thank you for the suggestion. A figure with annotation was added as suggested.

    1. “Data processing”, it is not necessary to detail each step for data processing in Visual 3D.

    We believe it is better to be detailed, for the sake of less experienced researchers.

    1. 3-2.5, it is suggested that the author could consider merging these sub-sessions.

    We assume that the reviewer means 2.3 – 2.5. Thank you for the suggestion. We merged the sections 2.3-2.4 as data processing and exporting are somewhat related. We merged 2.5 Reference Data with the subjects in section 2.1. “Reference and Subjects”.

    1. Results and discussion, both Results and Discussion should be separated parts, and in Results part the authors should present the main findings of this study while in Discussion part the authors should concentrate on the comparisons with previous related studies and further reveal the underlying mechanisms behind their findings.

    The discussion was separated and updated as suggested.

    1. Conclusion, this part should also be further strengthened based on the main findings of this study.

    We modified the conclusion according to the recommendation.

Reviewer 5 Report

Authors presented an interesting study to investigate the musculoskeletal morbidity following a transtibial amputation. While it is novel, there are still several issues should be properly addressed.

1. Introduction, please reorganize this part, removing the subtitles.

2. In terms of results, high resolution Figures should be added.

3. Line 245-264, please use academic writing, avoid using the subjective terms, such as 'we...' summarize, know, hypothesize... Please use objective expression.

Round 2

Reviewer 2 Report

bioengineering-1636384

Reviewer comments on the resubmission

In the resubmitted version of the manuscript, the authors have adequately responded concerning the topics mentioned in the initial round of reviewing.

  • English should be checked [i.e., at L130-131, L142, etc.].
  • It is suggested to add the part of the response of the 1st comment citing references 5-7 within the manuscript.
  • L130-131: What was the maximum number of trials conducted to record six ‘good’ trials?
  • L142: Provide citation for the ISB recommendation.
  • Figures 4+5: provide units for the horizontal axis.
  • L204-205, L237-245 & L259-276: Avoid including paragraphs containing a single sentence.
  • L240-241: Provide evidence/references for this statement.
  • References: check proper citation style for refs #19-21, #27-28.

Author Response

In the resubmitted version of the manuscript, the authors have adequately responded concerning the topics mentioned in the initial round of reviewing.

  • English should be checked [i.e., at L130-131, L142, etc.].

We updated the highlighted lines as well as minor language updates, as suggested.

  • It is suggested to add the part of the response of the 1st comment citing references 5-7 within the manuscript.

It was added as suggested as well.

  • L130-131: What was the maximum number of trials conducted to record six ‘good’ trials?

It was 13. The number was added to the manuscript.

  • L142: Provide citation for the ISB recommendation.

We provided the citation as recommended.

  • Figures 4+5: provide units for the horizontal axis.

Unit was added and legend was updated.

  • L204-205, L237-245 & L259-276: Avoid including paragraphs containing a single sentence.

The manuscript was updated accordingly.

  • L240-241: Provide evidence/references for this statement.

We provided 2 references and a small sentences with a third reference, as recommended.

  • References: check proper citation style for refs #19-21, #27-28.

We are not sure what the reviewer means with proper citation style. The manuscript references are managed by EndNote 20 using the "MDPI" style. However, let our reference manager update all references and refreshed the bibliography. Anyway, the citation should be revisited by the technical department should the manuscript be accepted for publication.

Finally, we are thankful for the time you spent to review our manuscript twice and for your sincere and constructive advices to make our manuscript better.

Reviewer 3 Report

The authors are to be congratulated for addressing the concerns of this reviewer. This manuscript provides a much better communication of the facets of the study.

Author Response

We are glad that the manuscript appeals to you now. Thank you for your constructive advices earlier.

Reviewer 4 Report

This study could show the distinction between reference normal persons and TTA patients, interesting topic. Revised document was quite good, I recommend to accept now. 

Author Response

(The authors gave the same response as above.)

Reviewer 5 Report

Authors made substantial revisions to improve the quality of this manuscript.

Author Response

(The authors gave the same response as above.)
